# Using nutritional survey data to inform the design of sugar-sweetened beverage taxes in low-resource contexts: a cross-sectional analysis based on data from an adult Caribbean population

Miriam Alvarado ![ORCID],[1] Rachel Harris ![ORCID],[2] Angela Rose ![ORCID],[3,4] Nigel Unwin ![ORCID],[5,6] Ian Hambleton ![ORCID],[3] Fumiaki Imamura ![ORCID],[7] Jean Adams ![ORCID] [1]

For numbered affiliations see end of article.

**Correspondence to**
Miriam Alvarado;
mra47@cam.ac.uk

## ABSTRACT

**Objective** Sugar-sweetened beverage (SSB) taxes have been implemented widely. We aimed to use a pre-existing nutritional survey data to inform SSB tax design by assessing: (1) baseline consumption of SSBs and SSB-derived free sugars, (2) the percentage of SSB-derived free sugars that would be covered by a tax and (3) the extent to which a tax would differentiate between high-sugar SSBs and low-sugar SSBs. We evaluated these three considerations using pre-existing nutritional survey data in a developing economy setting.

**Methods** We used data from a nationally representative cross-sectional survey in Barbados (2012–2013, prior to SSB tax implementation). Data were available on 334 adults (25–64 years) who completed two non-consecutive 24-hour dietary recalls. We estimated the prevalence of SSB consumption and its contribution to total energy intake, overall and stratified by taxable status. We assessed the percentage of SSB-derived free sugars subject to the tax and identified the consumption-weighted sugar concentration of SSBs, stratified by taxable status.

**Findings** Accounting for sampling probability, 88.8% of adults (95% CI 85.1 to 92.5) reported SSB consumption, with a geometric mean of 2.4 servings/day (±2 SD, 0.6, 9.2) among SSB consumers. Sixty percent (95% CI 54.6 to 65.4) of SSB-derived free sugars would be subject to the tax. The tax did not clearly differentiate between high-sugar beverages and low-sugar beverages.

**Conclusion** Given high SSB consumption, targeting SSBs was a sensible strategy in this setting. A substantial percentage of free sugars from SSBs were not covered by the tax, reducing possible health benefits. The considerations proposed here may help policymakers to design more effective SSB taxes.

### Strengths and limitations of this study

► A nationally representative dietary survey with two non-consecutive 24-hour dietary recalls allowed assessment of sugar-sweetened beverage (SSB) consumption patterns prior to the introduction of a tax on SSBs.

► Twenty-four-hour dietary recalls may be subject to reporting bias and may underestimate total SSB intake.

► Energy density (% of total energy intake) is reported to partially mitigate potential reporting biases.

► Data were not available on children, adolescents or adults over the age of 65 years.

► This is the first study that we are aware of to quantify the percentage of SSB-derived free sugars covered by a real-world SSB tax.

and consumption of SSBs is associated with a higher risk of diabetes, certain cancers and obesity.[2–10]

Given these health risks, the WHO and others have recommended taxing SSBs to reduce consumption.[11–15] A number of countries (including many small island developing states (SIDS) and low-income and middle-income countries) have introduced SSB taxes, at least in part, for health reasons.[12 16–19]

However, these taxes vary widely in design.[16] In some settings, taxable products have been narrowly defined, whereas elsewhere they have been defined to include all soft drinks (even those containing no or small amounts of free sugars).[16 20] These differences are likely to have important health implications.[21]

The design (or amendment) of SSB taxes should be informed by local consumption patterns as much as possible. Commercial purchase data (such as Nielsen and Kantar consumer panels) have been used to assess

## BACKGROUND

The WHO has recommended limiting free sugar consumption to less than 10% of total energy intake (TEI).[1] Free sugars include sugars added to food and beverages, as well as sugars in fruit juices.[1] Sugar-sweetened beverages (SSBs) are a major source of free sugars,

BMJ

SSB consumption patterns in the USA and the UK, but these data are costly and unavailable in some settings.[22] In lower-resource settings, in particular, it may be pragmatic to use pre-existing nutritional survey data to help inform context-specific policy design.[23 24] A recent review demonstrated that individual-level dietary surveys have been conducted in at least 116 countries, representing 88.7% of the global 2010 adult population.[23 24] These nutritional survey data may provide a feasible way to inform the design or amendment of SSB taxes across a variety of settings. We highlight three ways in which these data may be used to improve the design of SSB taxation.

First, there is great heterogeneity in SSB consumption levels worldwide.[25] We suggest that SSB taxes are more likely to be effective at substantially reducing free sugar consumption (in absolute terms) in settings in which SSB consumption levels are high and SSB-derived free sugars represent a high proportion of total energy intake.[26]

Second, we suggest that SSB taxes should cover a high proportion of regularly consumed SSBs, reducing substitution incentives.[12] If taxes are applied on a limited proportion of total SSBs consumed in a given population, the potential impact on health will be necessarily limited. If consumers substitute towards untaxed SSBs, health goals will be further undermined.

Finally, we suggest that SSB taxes should consistently differentiate between high-sugar and low-sugar products.[27 28] If SSB taxes are not consistently applied on all high-sugar SSBs, health goals will be further undermined especially if consumers substitute towards high-sugar untaxed SSBs.

Box 1 summarises these considerations.

### Case study: the Barbados SSB tax

The Government of Barbados implemented a 10% SSB tax in 2015.[18] Taxable products (both imported and locally manufactured) were defined according to the Harmonized System tariff classifications and included soda, juice drinks, energy and sports drinks (tariff headings 20.09 and 22.02).[18 29] Some SSBs were not included in the tax definition, such as sugar-sweetened drink mixes (eg, powdered juice and powdered hot chocolate) and sugar-sweetened syrups (eg, mauby, which is a local drink frequently sold as a syrup to be reconstituted at home).[18]

A nationally representative nutritional survey was conducted in 2012–2013, well in advance of the introduction of the Barbados SSB tax in 2015. We revisited these

data to assess the tax according to the considerations summarised in box 1. We aimed to assess the following questions: (1) What were pretax SSB consumption levels (in terms of volume and contribution to TEI)? (2) What percentage of SSB-related free sugars were covered by the tax? (3) Did the tax clearly differentiate between low and high-sugar beverages?

## METHODS
### Study Design & Population
We used nutritional survey data from Barbados, a country with a population of 293 131 (2018 estimate) and $18 600 GDP/capita (2017 estimate).[30] Barbados is likely to share characteristics with low/middle-income and other SIDS settings (limited access to commercial sales data, a product-based definition of taxable products, etc).

The data used in this study were from a subsample of the Health of the Nation study (HotN). The main HotN study was conducted between June 2012 and November 2013, with a response rate of 54% and the final sample size of 1234. Details of the overall sampling design, study recruitment and study procedures have been summarised elsewhere.[31 32] A subsample of 441 participants aged 25–64 years were randomly selected from the HotN study to complete two non-consecutive in-person 24-hour dietary recalls.[33] In total, 368 participants (83%) consented to participate in the substudy (for a combined response rate of 45%).

Each dietary recall was collected at home by a trained interviewer, using a standard multi-pass probing method, three-dimensional standardised food models and familiar measuring units.[34] Recalls were evenly distributed across quarters, with the exception of July–September when fewer recalls were conducted. The average time between the first and second recall was 6 days, and recalls were evenly distributed by day of the week. Data were processed using NutriBase Pro software.[35] Survey weights were used to reflect the clustered sampling design, to take into account the combined non-response rate and to match the age and sex distribution of the Barbados population as captured in the Barbados 2010 Census.[33]

We excluded participants with reported caloric intake less than 500 kcal/day or greater than 5000 kcal/day (n=5), those with missing covariate data (n=21), those with only one recall (n=1) and those with missing survey weights (n=7), leaving a total of 334 participants.

### Patient and public involvement
Participants were not involved in the design, conduct, reporting or dissemination of these analyses.

### Measures of SSB consumption
We estimated the prevalence of SSB consumption, defined as those with any reported SSB consumption on at least 1 day. Next, we estimated average volume consumed (mean SSB servings/day) among SSB consumers (excluding those who did not report any SSB

---

**Box 1    Proposed considerations to help inform design of sugar-sweetened beverage (SSB) taxes using pre-existing nutritional survey data**

1. Baseline levels of SSB consumption and contribution of SSB-derived free sugar to total energy intake.
2. Percentage of SSB consumption covered by SSB tax.
3. Extent to which SSB tax differentiates between high-sugar and low-sugar SSBs.

---

**Table 1** Consumption of SSBs among adults aged 25–64 years by demographic characteristics, Barbados 2012–2013: Barbados Salt Intake Study (n=334)

| | | Distribution (n=334) | Prevalence of any SSB consumption* (n=334) | | Volume (servings/day), given SSB consumption†‡ (n=300) | | TEI from SSB-derived free sugars, given SSB consumption†§ (n=300) | |
|---|---|---|---|---|---|---|---|---|
| | | % | % | 95% CI | Mean | Mean±2 SD | % | Mean±2 SD |
| Overall | Total | | 88.8 | 85.1 to 92.5 | 2.4 | 0.6, 9.2 | 9.2 | 2.1, 41.3 |
| By subgroup | | | | | | | | |
| Age | 25–44 | 51.1 | 89.1 | 83.7 to 94.6 | 2.7 | 0.9, 8.3 | 10.3 | 3.0, 35.6 |
| | 45–64 | 48.9 | 88.4 | 82.1 to 94.7 | 2.2 | 0.5, 9.8 | 8.2 | 1.5, 46.6 |
| Sex | Male | 48.8 | 89.7 | 83.7 to 95.7 | 2.8¶ | 0.9, 9.1 | 10.5¶ | 2.7, 41.3 |
| | Female | 51.2 | 87.9 | 83.0 to 92.9 | 2.1¶ | 0.5, 8.6 | 8.2¶ | 1.7, 39.6 |
| Education | <Tertiary | 62.9 | 90.9 | 85.7 to 96.2 | 2.7¶ | 0.8, 9.2 | 10.0¶ | 2.4, 41.3 |
| | Tertiary+ | 37.1 | 85.1 | 78.1 to 92.2 | 2.0¶ | 0.5, 8.4 | 8.0¶ | 1.6, 39.6 |

*Defined as >0 g of any SSB across two 24 hours recalls.
†Geometric means.
‡Defined as the mean volume (250 mL servings/day) from SSBs, among all SSB consumers. For estimates of 8 oz. per serving, each value is to be multiplied by 0.91.
§Defined as the percentage of TEI from SSB-derived free sugars, among all SSB-consumers.
¶Significant at p value <0.05 in survey-weighted bivariate logistic regression (prevalence of any SSB consumption models) or survey-weighted bivariate generalised linear regression with log-link function (volume, TEI models).
SSB, sugar-swetened beverages; TEI, total energy intake.

consumption). A serving was defined as 250 mL.[6] We reviewed each dietary recall and extracted product information for all reported SSBs.

Soft drinks were categorised based on whether they contained added sugars and whether they were subject to the Barbados SSB tax. Taxed SSBs included soda, juice drinks, energy/sports/malt drinks and other taxed SSBs; untaxed SSBs included sugar-sweetened powders (powdered juice drinks, hot chocolate), sugar-sweetened syrups (mauby), sweetened tea/coffee, sweetened condensed milk and other untaxed SSBs; and untaxed non-SSBs included water, no added sugar (NAS) fruit juice, milk, entirely artificially-sweetened beverages and other non-SSBs (see online supplementary appendix table 1).

We identified nutrient content for every beverage at the most detailed level possible (eg, brand and flavour). We relied on NutriBase nutrient content for international brands (and cross-checked these with local nutrient information panels for consistency). For brands not included in NutriBase, we collected nutrient information directly from product packaging and manufacturer websites (see online supplementary appendix text 1).

### Covariates

Demographic information and education history were collected at the first visit. We dichotomised age (25–44 years old, 45–64 years old) and education (secondary education or less compared with tertiary education, which included undergraduate, postgraduate and technical/vocational training).

### Statistical methods
#### Levels of SSB consumption

We estimated the prevalence of SSB consumption, and descriptive statistics (mean±1.96 SD) of levels of SSB consumption among consumers and the percentage of TEI from SSB-derived free sugars, stratified by covariates. Since SSB consumption was right-skewed (see online supplementary appendix figures 1 and 2), we report volume (in 250 mL servings) and percent of TEI using geometric means and SDs. To enable comparison with global estimates, we re-estimated overall SSB intake only using the arithmetic mean, including non-consumers and using 8 oz. (226.8 mL) as a serving size.[25]

#### Percentage of SSB-derived free sugars captured by tax

We re-estimated the prevalence of SSB consumption and percentage of TEI attributable to SSB-derived free sugars separately for taxed and untaxed SSBs. Then, we calculated the percentage of total SSB-derived free sugars subject to the tax. In calculating total SSB-derived free sugars, we excluded free sugars from non-SSBs (such as free sugars in NAS juice) and sugars naturally present in milk (which are not included in the definition of free sugars.)[1]

#### Free-sugar concentration

We estimated mean free-sugar concentration by SSB subcategory (ie, separately for sodas, SSB juice drinks, etc), weighted by reported consumption. Consumption-weighted estimates of free-sugar concentration were used to reflect consumption patterns (rather than reflecting the distribution of available free-sugars in the market). To

illustrate how nutritional survey data may be used to assess potential SSB tax tiers, we report mean per-person daily volume consumed by grams of free sugar per 100 mL.

All analyses were weighted by sampling probability and conducted using Stata V.14.0 (StataCorp LP, Texas, USA).

This study is reported according to the Strengthening Reporting of Observational Studies in Epidemiology Extension for Nutritional Epidemiology checklist (see online supplementary appendix table 2).[36]

## RESULTS
### Levels of SSB consumption
Eighty-eight percent of participants reported consuming SSBs at least once over the 2 days (table 1). Prevalence of SSB consumption did not differ significantly between subgroups. Among those who reported any consumption, mean per-person daily SSB intake was 2.4 250 mL servings (mean±2 SD, 0.6 9.2). To enable comparison with published estimates, we also report mean per-person daily SSB intake in 8 oz. servings across the whole study population (2.7 8 oz. servings (95% CI 2.5 to 2.9)). Men and those with less education reported consuming a higher volume of SSBs than their counterparts (p values of <0.001 and 0.004, respectively). TEI from SSB-related free sugars was 9.2% (mean±2 SD, 2.1, 41.3), with a similar patterning of results by subgroups.

### Percentage captured by tax
Seventy five percent of participants consumed taxed SSBs, and a similar percentage consumed untaxed SSBs (table 2). A higher percentage of men consumed taxed SSBs as compared with women (p=0.035). TEI attributable to taxed SSBs was 6.7% (mean±2 SD 1.7, 26.5), and TEI attributable to untaxed SSBs was 3.5% (mean±2 SD 0.4, 27.3). Those with less education consumed a higher percentage of TEI from taxed SSBs than those with higher education (p=0.01). Sixty-one percent of SSB-derived free sugars were taxed (95% CI 55.7 to 66.5), with no significant differences by subgroup.

### Free-sugar concentration
We estimated mean consumption-weighted free sugar concentration for each product category. As summarised in figure 1, sweetened condensed milk was associated with the highest concentration of free sugars (70 g/100 mL). Mauby, juice drinks and sodas had the next highest average free sugar concentrations. Five of the nine beverage types with more than 6.25 g free sugar/100 mL (Chile's SSB tax threshold) were untaxed. We also report mean per-person free sugar consumed (taking into account sugar concentration and consumption levels), by product type (see online supplementary appendix figure 3).

We assessed the mean per-person daily consumption of soft drinks (excluding those with free sugar <1 g/100 mL,

**Table 2** Prevalence of consumption and TEI (%) from SSB-derived free sugars among adults aged 25 to 64 years, stratified by subsequent taxable status, Barbados 2012–2013: Barbados Salt Intake Study (n=334)*

| | | Prevalence of any SSB Consumption† | | | | TEI from SSB-derived free sugars, given any SSB consumption‡§ | | | | SSB-derived free sugars from taxed SSBs¶ | |
| | | Taxed SSBs (n=334) | | Untaxed SSBs (n=334) | | Taxed SSBs (n=239) | | Untaxed SSBs (n=249) | | Percentage taxed (n=300) | |
| | | % | 95% CI | % | 95% CI | % | Mean±2 SD | % | Mean±2 SD | % | 95% CI |
|---|---|---|---|---|---|---|---|---|---|---|---|
| Overall | Total | 74.6 | 69.8 to 79.5 | 74.5 | 69.8 to 79.2 | 6.7 | 1.7, 26.5 | 3.5 | 0.4, 27.3 | 61.1 | 55.7 to 66.5 |
| By subgroup | | | | | | | | | | | |
| Age | 25–44 | 80.8 | 73.7 to 87.9 | 75.0 | 67.5 to 82.6 | 7.0 | 1.9, 25.7 | 3.6 | 0.6, 22.6 | 64.2 | 58.3 to 70.1 |
| | 45–64 | 68.1 | 59.5 to 76.8 | 74.0 | 65.5 to 82.4 | 6.3 | 1.4, 27.3 | 3.4 | 0.3, 33.3 | 57.0 | 47.5 to 66.5 |
| Sex | Male | 79.8** | 72.8 to 86.8 | 70.8 | 62.3 to 79.3 | 7.2 | 1.9, 27.9 | 3.9 | 0.5, 30.4 | 62.1 | 56.0 to 68.2 |
| | Female | 69.7** | 63.1 to 76.3 | 78.0 | 72.9 to 83.2 | 6.1 | 1.5, 24.6 | 3.2 | 0.4, 24.1 | 59.6 | 51.5 to 67.7 |
| Education | <Tertiary | 77.4 | 70.5 to 84.3 | 76.8 | 70.2 to 83.5 | 7.2** | 1.8, 28.6 | 3.5 | 0.5, 26.9 | 63.2 | 57.7 to 68.7 |
| | Tertiary+ | 69.9 | 61.8 to 78.0 | 70.6 | 63.5 to 77.7 | 5.7** | 1.5, 21.8 | 3.4 | 0.4, 28.0 | 56.7 | 48.1 to 65.3 |

*The tax was introduced in 2015, so we retrospectively apply the definition of taxable goods to consumption data reported from 2012 to 2013.
†Defined as >0gr of taxed/untaxed SSBs across two 24-hour recalls
‡Geometric means.
§Defined as the mean TEI from SSB-derived free sugars divided by TEI, among all taxed and untaxed SSB-consumers separately.
¶Defined as the percentage of SSB-derived free sugars that were included in the original Barbados SSB tax definition of taxable products, among all SSB-consumers.
**Significant at p value <0.05 in survey-weighted bivariate logistic regression (prevalence of any SSB consumption models) or bivariate generalised linear regression with log-link function (TEI).
SSB, sugar-sweetened beverage; TEI, total energy intake.

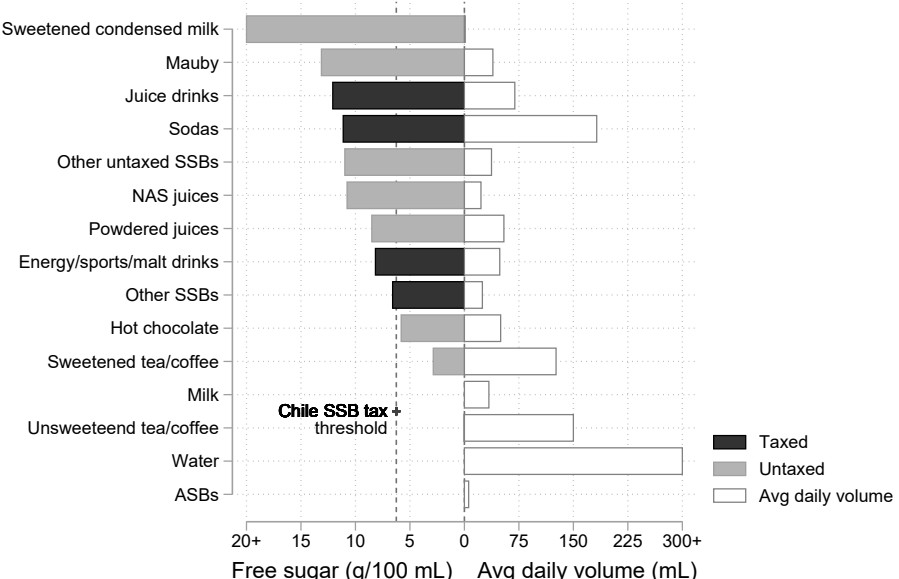

**Figure 1** Mean consumption-weighted free sugar concentration by product type (g/100 mL) stratified by subsequent taxable status, and mean per-person daily volume consumed (mL) in Barbados 2012–2013: Barbados Salt Intake Study (n=334). We present NAS juice sugars for comparison in the figure and include a dashed line to represent the SSB tax threshold used in Chile (6.25 g sugar/100 mL).[1 43] The tax was introduced in 2015, and we retrospectively apply the definition of taxable goods to consumption data reported from 2012 to 2013. NAS, no added sugar; SSB, sugar-sweetened beverage.

and including home-prepared SSBs and no added sugar juice) by free sugar concentration (figure 2), stratified by taxed/untaxed SSBs. Nearly half of the drinks consumed with the highest free sugar levels (12+ g/100 mL) were not subject to the tax (see online supplementary appendix table 3 and text 2 for examples of the products in each category by free sugar concentration).

## DISCUSSION

We used pre-existing nutritional survey data to assess three important considerations around the introduction and design of the Barbados SSB tax.

SSB consumption levels among adults aged 25–64 years in Barbados were very high (2.7 8 oz. servings/day, 95% CI 2.5 to 2.9) compared with global estimates (0.58

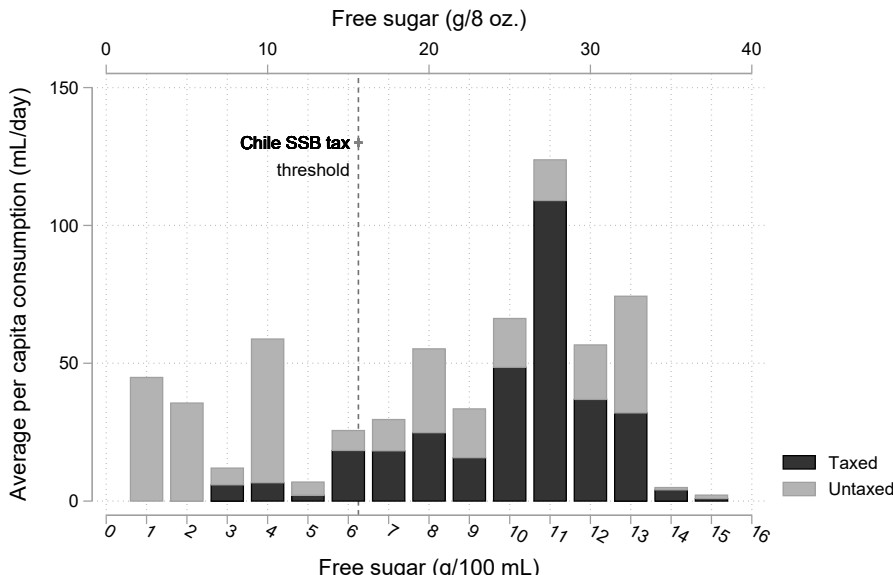

**Figure 2** Mean per-person daily volume consumed (mL) by free sugar concentration (g/100 mL and g/8 oz.), stratified by subsequent taxable status in Barbados 2012–2013: Barbados Salt Intake Study (n=334). The tax was introduced in 2015, and we retrospectively apply the definition of taxable goods to consumption data reported from 2012 to 2013.

8 oz. servings/day, 95% CI 0.37 to 0.83), suggesting that interventions to reduce SSB consumption in Barbados had the potential to reduce absolute free sugar consumption more than in settings with low baseline consumption.[25] SSB-derived free sugar accounted for 9.2% of TEI (mean±2SD 2.1, 41.3), and therefore nearly half of the population exceeded the WHO's recommendation for total free sugar (10%, including sweets, jams, confectionery, etc) solely from SSB consumption.[1]

The Barbados SSB tax captured a moderate percentage of SSB-derived free sugars (61.1%, 95% CI 55.7 to 66.5), possibly incentivising substitution to untaxed SSBs and dampening the potential health impact of the tax.

The Barbados SSB tax did not clearly differentiate between consumption-weighted high-sugar and low-sugar products, which may further incentivise substitution to high-sugar untaxed alternatives in particular.

### Strengths and limitations

The considerations assessed here reflect some aspects of SSB tax design, but many other context-specific factors need to be considered (eg, tax level, tax structure, availability of alternative beverages, public acceptability, market structure, revenue ear-marking, related policies, etc). However, this assessment illustrated important aspects of context-specific consumption patterns and may provide useful information to policymakers.

Given the data available, we were not able to assess SSB consumption patterns among children, young adults or adults over 65 years. The combined response rate was 45%, comparable to that of a similar national dietary survey in the UK (47%).[37] Survey weights were used to take the population representativeness into account as much as possible and to match the age and sex distribution of the Barbados population. However, if participants differed systematically from non-participants in ways not accounted for by the survey weights, our estimates of SSB consumption may not be representative of the broader population. There was a dip in recalls conducted between July and September, suggesting that recall data may be slightly seasonally biased. July–September represent the hottest months in Barbados, and SSB consumption may increase during these months, which would imply that we may have underestimated consumption.[38] Underestimation could have also occurred because of the subjectivity in the two 24-hour recall data, which may have been partially mitigated by the energy density approach (% of TEI).

### In relation to other studies

The Global Burden of Disease (GBD) 2010 study estimated that SSB consumption in Barbados was between 2.0 and 2.4 8 oz. servings/day, lower than our comparable estimate of 2.7 servings/day.[25] This difference may reflect that the GBD estimate for Barbados was derived from a study conducted in Jamaica between 1993 and 1995[39] and an unpublished analysis.[25]

In comparison to national measures of SSB consumption from other settings, our estimates were relatively high. Han and Powell estimated the 2-day prevalence of SSB consumption among US adults was 50%, lower than our comparable estimate of 89% among adults in Barbados.[40] A study of Dutch adults found that SSBs and non-SSBs accounted for 5.1% of TEI and a study of Australian children estimated an SSB contribution of 4.4%, much lower than our 9.2% estimate.[2 41 42]

This is the first study that we are aware of to quantify the percentage of SSB-derived free sugars covered by an SSB tax. Given heterogeneous SSB consumption worldwide, it would be valuable to repeat this approach in different settings to assess both the potential (in general) of an SSB tax to target sources of soft drink-derived free sugar, as well as to evaluate the specific definition of proposed future taxes. Powell et al have assessed the distribution of sugar concentration by consumption of ready-to-drink SSBs (excluding home-prepared SSBs) in the USA and identified two clusters of highly consumed concentration levels.[28] They recommended that SSB tax thresholds should be set at 5 g/8 oz. below these highly consumed clusters to encourage reformulation.[28] This guidance would imply a threshold of around 8 g/100 mL given the distribution we observed in Barbados, somewhat higher than the threshold used in Chile (6.25 g/100 mL).[43] More empirical work is needed to understand how companies respond to these thresholds in practice and to assess how home-prepared SSBs compare in terms of sugar concentration levels in other settings.

### Meaning of the study
#### Implications for Barbados

Adult SSB consumption levels were high before the introduction of the Barbados SSB tax. However, the definition of taxable products suggests that the tax was only likely to cover a moderate proportion of SSB-related free sugar consumption. While the Barbados tax was amended in 2017 to include store-bought mauby syrup (tariff heading 21.06), homemade mauby and other homemade SSBs remain difficult to address through a tax.[44] To maximise health benefit, the tax could be further amended to cover a higher proportion of SSB-derived free sugars, such as powdered juice drinks and powdered hot chocolate.

Some untaxed products (eg, no added sugar juices and powdered juices) contain higher levels of free sugars than taxed products, suggesting that substitution to untaxed beverages could have the unintended consequence of increasing sugar consumption. To further maximise health benefit, the tax could be amended to include some of these products. For example, including no added sugar juices in the SSB tax may further help to deter free sugar consumption.[45–47] Recent dietary guidelines in Barbados suggest limiting no added sugar juice intake to 250 mL/day, and similar guidelines in the UK recommend a threshold of less than 150 mL/day. In addition, different tax designs may be considered, such as basing the tax on

sugar content or introducing sugar-content based tiers, as has been done elsewhere.[20]

## Implications for other settings

We found that it was feasible to use pre-existing nutritional survey data to assess these considerations, and suggest that they may usefully inform SSB tax design.

In countries which use tariff headings as the basis for SSB taxation (eg, Barbados, St. Kitts and Nevis, Bolivia and South Africa), the tariff headings selected for taxation may vary substantially.[48–50] For example, in South Africa taxable tariff headings included 18.06 ('cocoa powder… for making beverages'), 19.01 ('malt extract… for making beverages'), 21.06 ('syrups and other concentrates or preparation for making beverages') and 22.02 ('waters…containing added sugar…'), while in Barbados taxable tariff headings (in the original law) included 20.09 (' Fruit juices … and vegetable juices…') and 22.02 ('waters…containing added sugar…') and the 2017 amendment included 21.06 ('mauby syrup and other flavoured or coloured sugar syrups').[18 44 50] When SSB taxes are defined by tariff headings or other types of product categories, care should be taken that all high-sugar products are taxed to limit incentives for substitution.

A potential limitation of SSB taxes, in general, is that they do not cover home-prepared SSBs. In contexts where a high absolute volume of SSBs are home-prepared, an SSB tax has less health potential irrespective of the definition of taxable products. Complementary mass media or education campaigns that target untaxed sources of SSB-derived free sugars may be helpful in addressing free sugar consumption overall, given the limitations of any tax to capture all these beverages.

It was feasible to use existing nutritional survey data to assess several important considerations around SSB taxation, and these data offered some advantages over other potential data sources. Nutritional survey data can provide insight around homemade and on-the-go SSB consumption, although they may be limited by small sample sizes (which may preclude subgroup analyses) and infrequent administration. Nevertheless, standard nutritional surveys, when combined with detailed nutrient content data, can provide an opportunity to assess consumption patterns and highlight opportunities to design tailored context-informed SSB taxes.

## CONCLUSION

We used nutritional survey data to demonstrate high levels of SSB consumption (both in volume and as a percentage of total energy intake) among adults in Barbados prior to the introduction of the Barbados SSB tax. The Barbados SSB tax could be amended to apply to additional SSB products, potentially increasing possible health benefits. SSB taxes may miss home-prepared SSBs, and additional interventions may be needed to address these sources of free-sugars. Evaluating these considerations (baseline SSB consumption levels, the percentage of all SSBs that would be taxed, and the ability of a tax to differentiate between high-sugar and low-sugar soft drinks) in other settings may help to improve SSB tax design and increase potential positive health impacts.

**Author affiliations**
[1]Centre for Diet and Activity Research, MRC Epidemiology Unit, University of Cambridge, Cambridge, UK
[2]Faculty of Medical Sciences, University of the West Indies, Cave Hill, Barbados
[3]George Alleyne Chronic Disease Research Centre, Caribbean Institute for Health Research, University of the West Indies at Cave Hill, Bridgetown, Barbados
[4]Epidemiology Department, Epiconcept, Paris, France
[5]Global Diet and Activity Research, MRC Epidemiology Unit, University of Cambridge, Cambridge, UK
[6]College of Medicine and Health, University of Exeter, Truro, UK
[7]MRC Epidemiology Unit, University of Cambridge, Cambridge, UK

**Acknowledgements** The full Barbados SSB Tax Evaluation Group is: Trevor Hassell (Chairperson of the group, National NCD Commission of Barbados and Healthy Caribbean Coalition); Miriam Alvarado (Centre for Diet and Activity Research (CEDAR), University of Cambridge); Kenneth George (Ministry of Health), Ian Hambleton (UWI); Maisha Hutton (Healthy Caribbean Coalition); Winston Moore (UWI); Madhuvanti Murphy (UWI); Alafia Samuels (UWI); Godfrey Xuereb (PAHO); Nigel Unwin (UWI; CEDAR, University of Cambridge); Cyril Gill (Ministry of Finance) and Arthur Phillips (Ministry of Health).

**Contributors** MA was involved in conceptualising the study, developing the methodology, analysing the data and writing and revising the manuscript. NU was involved in conceptualisation, funding acquisition, developing the methodology, and reviewing and editing the manuscript. JA was involved in conceptualisation, developing the methodology, and reviewing and editing the manuscript. RH was involved in data curation and data quality checks, developing the methodology and reviewing and editing the manuscript. AR was involved in data curation and software development, developing the methodology, validating results and reviewing and editing the manuscript. IH was involved in data curation, developing the methodology, reviewing data visualisations, and reviewing and editing the manuscript. FI was involved in developing the methodology, reviewing data presentation, and reviewing and editing the manuscript. All authors read and approved the final manuscript.

**Funding** The group is supported by a Technical Advisory Committee: Jean Adams (CEDAR, University of Cambridge); Deliana Kostova (US Center for Disease Control and Prevention); Marc Suhrcke (University of York); Anne Marie Thow (University of Sydney)Nita Forouhi generously offered her time to review a preliminary statistical analysis plan and Dave Collins provided technical expertise around survey-based estimates of SSB consumption. The overall evaluation received support from the Canadian International Development Research Centre (grant no. 1 07 604–001) and from the US Center for Disease Control and Prevention (TEPHINET). MA is funded through a Gates Cambridge PhD Scholarship, and received travel funding from Robinson College, the Global Food Security Fund, the Luca D'Agliano Scholarship, the Yates Unilever Fund and the Smuts Memorial Fund. JA is funded by the Centre for Diet and Activity Research (CEDAR), a UKCRC Public Health Research Centre of Excellence. FI was funded by the Medical Research Council Epidemiology Unit Core Support (MC_UU_12015/5). Funding from the British Heart Foundation, Cancer Research UK, Economic and Social Research Council, Medical Research Council, the National Institute for Health Research, and the Wellcome Trust, under the auspices of the UK Clinical Research Collaboration, is gratefully acknowledged. The Health of the Nation Study and the Barbados Salt Intake Study received funding from The Ministry of Health of the Government of Barbados.

**Competing interests** None declared.

**Patient and public involvement** Patients and/or the public were not involved in the design, or conduct, or reporting, or dissemination plans of this research.

**Patient consent for publication** Not required.

**Ethics approval** Ethics approval was given by the University of the West Indies Cavehill Institutional Review Board.

**Provenance and peer review** Not commissioned; externally peer reviewed.

**Data availability statement** Data are available upon reasonable request. The datasets analysed during the current study (deidentified participant data) are available from the corresponding author (0000-0003-2864-9410) on reasonable request.

**ORCID iDs**
Miriam Alvarado http://orcid.org/0000-0003-2864-9410
Rachel Harris http://orcid.org/0000-0002-5205-4989
Angela Rose http://orcid.org/0000-0002-2493-8082
Nigel Unwin http://orcid.org/0000-0002-1368-1648
Ian Hambleton http://orcid.org/0000-0002-5638-9794
Fumiaki Imamura http://orcid.org/0000-0002-6841-8396
Jean Adams http://orcid.org/0000-0002-5733-7830

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
