## [Reviewer comments · BMJ Open]

ARTICLE DETAILS

TITLE (PROVISIONAL)	Using nutritional survey data to inform the design of sugar-sweetened beverage taxes in low-resource contexts: a cross-sectional analysis based on data from an adult Caribbean population
AUTHORS	Alvarado, Miriam; Harris, Rachel; Rose, Angela; Unwin, Nigel; Hambleton, Ian; Imamura, Fumiaki; Adams, J

VERSION 1 – REVIEW

REVIEWER	Yen-Han Lee Department of Applied Health Sciences, School of Public Health, Indiana University Bloomington, IN 47405, U.S.A.
REVIEW RETURNED	04-Jan-2020

GENERAL COMMENTS	This study used nutritional survey data to investigate the design of sugar sweetened beverage taxes in low-resource contexts. This is an important topic, given that sugar sweetened beverage tax is a critical way to reduce the consumption of sweetened beverage. Using Barbados as the study subject is very unique in its own way. In general, this is also a very strong manuscript with clear tables and figures. I am providing some suggestions here that I hope to help with authors' revisions. First of all, the introduction section is somewhat fragmented. I would encourage the authors to move things around and make the flow better. In addition, the authors clearly explain the study design and sample. However, I do have a concern. It seems to me that a response rate of 54% is somewhat low. Do authors figure out the factors affecting the low response rate? The final study sample includes 334 adults, but the original study sample includes 1,234 participants. Can authors provide more information about the discrepancy? On page 8 (lines 144 to 146), the authors clarify the selection criteria, but those missing/excluded values do not add up with the discrepancy between the original sample size and the final study sample. This should be carefully addressed. Thank you.
--

REVIEWER	Andrea Teng University of Otago
REVIEW RETURNED	04-Mar-2020

GENERAL COMMENTS	This study presents a framework for informing SSB tax design, and then assesses the 2015 Barbados 10% SSB tax against this framework for (1) baseline consumption (including prevalence, number of servings and total energy intake as a proportion of recommended), (2) the proportion of SSB free sugar covered by
--

	the taxes, and (3) how well the tax distinguished between high and low sugar SSBs. It appears to be the first study to quantify SSB free sugars covered by a real world SSB tax using national representative nutrition survey data. Well structured, well written and makes an interesting contribution. The study makes a convincing argument that a high proportion of SSBs should be covered by a SSB tax (part two of framework) to improve potential health impacts. The data here present a unique way of assessing this, and highlights the lack of comprehensiveness of the Barbados tax. This is relevant to other LMICs and SIDS using similar criteria for SSB tax, and it would be good to have more details about the Barbados criteria that were used, and how they specifically could be improved eg, what harmonised codes did the include and which ones did they not? If this study was aiming to provide comprehensive criteria for strong tax design, there are several other factors that would be included in the framework such as size of the tax, structure of the tax (nutrient-specific, volumetric or ad valorem), availability of alternative beverages, hypothecation/other NCD policies etc. Although this point is touched on (Line 234) the framework appears to over promise and seems to have been created based on what was possible to assess from the available nutrition data. The paper could be better framed as an evaluation of SSB free sugar coverage of the Barbados SSB tax and an evaluation of one or two scenarios that might have improved the coverage of the tax. Eg, applying the tax to wider number of SSB categories, use of a tax that differs by sugar concentration thresholds, use of nutrient-specific taxes based on sugar concentration. It is unclear to me how part one of the framework assessing the level of SSB baseline consumption, can improve tax design as claimed, although I accept it may be useful background. More justification is needed. Part three of the framework about distinguishing between high and low sugar SSBs is interesting for future tax design and impact of proposed thresholds, but how could the Barbados tax be changed to improve the way it distinguishes between high and low sugar SSBs? Other comments Title: could mention Barbados rather than Caribbean to be more specific Line 58: could this be presented as a confidence interval? Line 94; specify that this in absolute terms (is it not possible relative declines may be similar irrespective of how high consumption was to begin with?) Line 249; shouldn't this be lower rather than higher? Line 239; What were the possible impacts of combined response rate of 45% on study results? and a further 9% of participants excluded /missing data?
--	--

VERSION 1 – AUTHOR RESPONSE

Reviewer: 1

Reviewer Name: Yen-Han Lee

Institution and Country: Department of Applied Health Sciences, School of Public Health, Indiana University Bloomington, IN 47405, U.S.A.

Please state any competing interests or state 'None declared': None declared.

Please leave your comments for the authors below

This study used nutritional survey data to investigate the design of sugar sweetened beverage taxes in low-resource contexts. This is an important topic, given that sugar sweetened beverage tax is a critical way to reduce the consumption of sweetened beverage. Using Barbados as the study subject is very unique in its own way. In general, this is also a very strong manuscript with clear tables and figures. I am providing some suggestions here that I hope to help with authors' revisions.

Thank you for your kind feedback.

First of all, the introduction section is somewhat fragmented. I would encourage the authors to move things around and make the flow better.

Thank you for this feedback. We have made a number of changes with the aim of improving the flow of the introduction. Specifically, we re-structured Lines 89-109, including shifting the paragraph about commercial data availability to lines 90-97 and adding additional paragraph breaks in Lines 89 and 109. We hope this has improved the flow of this section.

In addition, the authors clearly explain the study design and sample. However, I do have a concern. It seems to me that a response rate of 54% is somewhat low. Do authors figure out the factors affecting the low response rate?

For overall HotN study recruitment, the response rate was heavily influenced by a number of households who refused to give permission to be contacted by the study team initially (n=491). These households were initially approached by the Barbados Continuous Labour Force Sample Survey (CLFSS) during three quarters of 2011.

Of those households which did consent for their details to be passed on to the HotN study team, 140 households refused to participate after being contacted by the HotN study team, 210 individuals within households refused to participate once selected and contact with 52 selected individuals was not successful. Finally 150 study appointments were missed (defined as two or more missed appointments) and categorized as refusal to participate. In total, 1234 participants took part in the survey, yielding a response rate of $1234/(1234+491+140+210+150+52)$, or 54.2%. We have added an additional reference (Unwin N, Rose AMC, George KS, Hambleton IR, Howitt C. The Barbados Health of the Nation Survey: Core Findings. Chronic Disease Research Centre, The University of the West Indies and the Barbados Ministry of Health: St Michael, Barbados, January 2015) which includes these details (pages 12-14).

We agree that a higher response rate would have been desirable, and acknowledge the limitations for generalizability to the broader adult Barbados population. We expand on this as a potential limitation in the discussion (Lines 239-244):

The combined response rate was 45%, comparable to that of a similar national dietary survey in the United Kingdom (47%)[36]. Survey weights were used to take the population representativeness into account as much as possible and to match the age and sex distribution of the Barbados population. However, if participants differed systematically from non-participants in ways not accounted for by the survey weights, our estimates of SSB consumption may not be representative of the broader population.

An attempt was made to collect demographic data on all non-responding households, and this was achieved for 50% of non-responding households. Using these data, the mean age of non-responders was slightly lower (49.9 years compared to 51 years) and non-responders were more likely to be male (50% compared to 38%). However, given the missing information on 50% of non-responders, it was

not possible to fully assess any differences between participants and non-participants.

The final study sample includes 334 adults, but the original study sample includes 1,234 participants. Can authors provide more information about the discrepancy? On page 8 (lines 144 to 146), the authors clarify the selection criteria, but those missing/excluded values do not add up with the discrepancy between the original sample size and the final study sample. This should be carefully addressed.

Thank you for highlighting the need to clarify this point. We have modified the following text to explain this more thoroughly.

Lines 128-133:

The data used in this study were from a sub-sample of the Health of the Nation study (HotN). The main HotN study was conducted between June 2012 and November 2013, with a response rate of 54% and final sample size of 1,234. Details of the overall sampling design, study recruitment and study procedures have been summarized elsewhere. A sub-sample of 441 participants aged 25 to 64 were randomly selected from the HotN study to complete two non-consecutive in-person 24-hour dietary recalls [32]. In total, 368 participants (83%) consented to participate in the sub-study (for a combined response rate of 45%).

We agree that it was difficult to track these numbers in the original manuscript. We hope this is now more clear: 368 participants consented to participate in the sub-study, and after excluding those with excessively high/low reported intake (5), missing covariate data (21), only one recall (1) and missing survey weights (7, for a total of 34 exclusions), we reached a total of 334 participants. Many thanks for suggesting this clarification.

Reviewer: 2

Reviewer Name: Andrea Teng

Institution and Country: University of Otago

Please state any competing interests or state 'None declared': None declared

Please leave your comments for the authors below

This study presents a framework for informing SSB tax design, and then assesses the 2015 Barbados 10% SSB tax against this framework for (1) baseline consumption (including prevalence, number of servings and total energy intake as a proportion of recommended), (2) the proportion of SSB free sugar covered by the taxes, and (3) how well the tax distinguished between high and low sugar SSBs. It appears to be the first study to quantify SSB free sugars covered by a real world SSB tax using national representative nutrition survey data. Well structured, well written and makes an interesting contribution.

Thank you for your kind feedback.

The study makes a convincing argument that a high proportion of SSBs should be covered by a SSB tax (part two of framework) to improve potential health impacts. The data here present a unique way of assessing this, and highlights the lack of comprehensiveness of the Barbados tax. This is relevant to other LMICs and SIDS using similar criteria for SSB tax, and it would be good to have more details about the Barbados criteria that were used, and how they specifically could be improved eg, what harmonised codes did the include and which ones did they not?

Thank you for highlighting the importance of clarifying the specific harmonised tariff codes that were

used, we agree this is a key aspect of the manuscript. We have added the following text:

Line 113:

“(tariff headings 20.09 and 22.02)”

Lines 291-297:

“In countries which use tariff headings as the basis for SSB taxation (e.g. Barbados, St. Kitts and Nevis, Bolivia and South Africa), the tariff headings selected for taxation may vary substantially[48–50]. For example, in South Africa taxable tariff headings included 18.06 (“cocoa powder... for making beverages”), 19.01 (“malt extract... for making beverages”), 21.06 (“syrops and other concentrates or preparation for making beverages”) and 22.02 (“waters...containing added sugar...”), while in Barbados taxable tariff headings (in the original law) included 20.09 (“Fruit juices ... and vegetable juices...”) and 22.02 (“waters...containing added sugar...”) and the 2017 amendment included 21.06 (“mauby syrup and other flavoured or coloured sugar syrups”).[18,44,50]”

If this study was aiming to provide comprehensive criteria for strong tax design, there are several other factors that would be included in the framework such as size of the tax, structure of the tax (nutrient-specific, volumetric or ad valorem), availability of alternative beverages, hypothecation/other NCD policies etc. Although this point is touched on (Line 234) the framework appears to over promise and seems to have been created based on what was possible to assess from the available nutrition data. The paper could be better framed as an evaluation of SSB free sugar coverage of the Barbados SSB tax and an evaluation of one or two scenarios that might have improved the coverage of the tax. Eg, applying the tax to wider number of SSB categories, use of a tax that differs by sugar concentration thresholds, use of nutrient-specific taxes based on sugar concentration.

We agree with you and have made the following changes to re-frame the manuscript in line with your suggestions:

Re-framing Box 1 with a modified title (“Proposed considerations to help inform design of sugar-sweetened beverage (SSB) taxes using pre-existing nutritional survey data”) and updating the numbered points to highlight which aspects should be assessed (rather than framing these as criteria that should be met).

1. Baseline levels of SSB consumption and contribution of SSB-derived free sugar to total energy intake (TEI)
2. Percentage of SSB consumption covered by SSB tax
3. Extent to which SSB tax differentiates between high- and low-sugar SSBs

Editing the abstract to now read (Lines 45-49):

“We aimed to use a pre-existing nutritional survey data to inform SSB tax design by assessing: 1) baseline consumption of SSBs and SSB-derived free sugars, 2) the percentage of SSB-derived free sugars that would be covered by a tax, and 3) the extent to which a tax would differentiate between high- and low-sugar SSBs. We evaluated these three considerations using pre-existing nutritional survey data in a developing economy setting.”

Referring to “considerations” rather than “criteria” throughout

Editing sections of the Introduction, Discussion and Conclusion in line with this re-framing:

Lines 95-97:

These nutritional survey data may provide a feasible way to inform the design or amendment of SSB taxes across a variety of settings. We highlight three ways in which these data may be used to improve the design of SSB taxation.

Lines 218-219:

We used pre-existing nutritional survey data to assess three important considerations around the introduction and design of the Barbados SSB tax.

Lines 233-235:

The considerations assessed here reflect some aspects of SSB tax design, but many other context-specific factors need to be considered (e.g. tax level, tax structure, availability of alternative beverages, public acceptability, market structure, revenue ear-marking, related policies, etc.).

Lines 288-289:

We found that it was feasible to use pre-existing nutritional survey data to assess these considerations, and suggest that they may usefully inform SSB tax design.

Lines 305-306:

It was feasible to use existing nutritional survey data to assess several important considerations around SSB taxation, and these data offered some advantages over other potential data sources.

Lines 317-320:

Evaluating these considerations (baseline SSB consumption levels, the percentage of all SSBs that would be taxed, and the ability of a tax to differentiate between high- and low-sugar soft drinks) in other settings may help to improve SSB tax design and increase potential positive health impacts.

It is unclear to me how part one of the framework assessing the level of SSB baseline consumption, can improve tax design as claimed, although I accept it may be useful background. More justification is needed.

Thank you for raising this point. Our intention with part one of the framework was to suggest that taxing SSBs is most likely to have an impact on health in contexts in which SSBs already (at baseline) contribute substantially to total caloric intake and free sugar consumption. We have attempted to clarify this in the text:

Lines 98-101:

First, there is great heterogeneity in SSB consumption levels worldwide[25]. We suggest that SSB taxes are more likely to be effective at substantially reducing free sugar consumption (in absolute terms) in settings in which SSB consumption levels are high and SSB-derived free sugars represent a high proportion of total energy intake[26].

Part three of the framework about distinguishing between high and low sugar SSBs is interesting for future tax design and impact of proposed thresholds, but how could the Barbados tax be changed to improve the way it distinguishes between high and low sugar SSBs?

The Barbados tax could be amended in several ways to distinguish between high and low-sugar SSBs. For instance, we have clarified this in the text:

Lines 281-286:

To further maximize health benefit, the tax could be amended to include some of these products. For example, including no added sugar juices in the SSB tax may further help to deter free sugar

consumption[45–47]. Recent dietary guidelines in Barbados suggest limiting no added sugar juice intake to 250 ml/day, and similar guidelines in the UK recommend a threshold of less than 150ml/day. In addition, different tax designs may be considered, such as basing the tax on sugar content or introducing sugar-content based tiers, as has been done elsewhere.[20]

Other comments

Title: could mention Barbados rather than Caribbean to be more specific

We elected to use 'Caribbean' to highlight that we think this analysis may be of particular relevance to other Caribbean countries, many of which share similar patterns of SSB consumption and/or have introduced SSB taxes with important similarities to the Barbados SSB tax.

This follows convention, with other papers drawing on Barbados-specific data referring to 'Caribbean' populations in their titles (e.g. "A cross-sectional study of physical activity and sedentary behaviours in a Caribbean population: combining objective and questionnaire data to guide future interventions" and "Sodium and potassium excretion in an adult Caribbean population of African descent with a high burden of cardiovascular disease.")

Line 58: could this be presented as a confidence interval?

Since we present this result as a geometric mean to better reflect the underlying distribution of the variable of interest, we chose to report this measure of variability in the abstract, in line with the measures we present in Table 1. If we presented "confidence intervals" instead, we worry it may become unclear (and misleading) whether the measure captures uncertainty of the mean or variability of the exposure distribution (i.e. $\text{mean} \pm 1.96 \times \text{SE}$ or $\text{mean} \pm 2.0 \times \text{SD}$). To avoid any confusion, we would like to keep $\text{geometric mean} \pm 2 \times \text{SD}$ in our presentation. We agree it is difficult to make this clear in the limited word count of the abstract, but hope to clarify this choice in the manuscript text, i.e. Lines 170-171:

Since SSB consumption was right-skewed (see Appendix Figures 1 and 2), we report volume (in 250 mL servings) and percent of TEI using geometric means and SDs.

Line 94; specify that this in absolute terms (is it not possible relative declines may be similar irrespective of how high consumption was to begin with?)

We have removed this line, and clarified in a related line that we are referring to absolute declines. Thank you for this clarification.

Lines 98-101:

We suggest that SSB taxes are more likely to be effective at substantially reducing free sugar consumption (in absolute terms) in settings in which SSB consumption levels are high and SSB-derived free sugars represent a high proportion of total energy intake[26].

Line 249; shouldn't this be lower rather than higher?

Yes, thanks very much for making this correction. We have amended the text accordingly (Line 251).

Line 239; What were the possible impacts of combined response rate of 45% on study results? and a further 9% of participants excluded /missing data?

We have added the following text to clarify the potential implications of this, thank you for highlighting this.

Lines 242-244:

However, if participants differed systematically from non-participants in ways not accounted for by the

survey weights, our estimates of SSB consumption may not be representative of the broader population.